# Structures and function of a tailoring oxidase in complex with a nonribosomal peptide synthetase module

Camille Marie Fortinez[1,2], Kristjan Bloudoff[1,2], Connor Harrigan[1,2], Itai Sharon [1,2], Mike Strauss [2,3] &
T. Martin Schmeing [1,2✉]

Nonribosomal peptide synthetases (NRPSs) are large modular enzymes that synthesize secondary metabolites and natural product therapeutics. Most NRPS biosynthetic pathways include an NRPS and additional proteins that introduce chemical modifications before, during or after assembly-line synthesis. The bacillamide biosynthetic pathway is a common, three-protein system, with a decarboxylase that prepares an NRPS substrate, an NRPS, and an oxidase. Here, the pathway is reconstituted in vitro. The oxidase is shown to perform dehydrogenation of the thiazoline in the peptide intermediate while it is covalently attached to the NRPS, as the penultimate step in bacillamide D synthesis. Structural analysis of the oxidase reveals a dimeric, two-lobed architecture with a remnant RiPP recognition element and a dramatic wrapping loop. The oxidase forms a stable complex with the NRPS and dimerizes it. We visualized co-complexes of the oxidase bound to the elongation module of the NRPS using X-ray crystallography and cryo-EM. The three active sites (for adenylation, condensation/cyclization, and oxidation) form an elegant arc to facilitate substrate delivery. The structures enabled a proof-of-principle bioengineering experiment in which the BmdC oxidase domain is embedded into the NRPS.

[1] Department of Biochemistry, McGill University, Montréal, QC H3G 0B1, Canada. [2] Centre de recherche en biologie structurale, McGill University, Montréal, QC H3G 0B1, Canada. [3] Department of Anatomy and Cell Biology, McGill University, Montréal, QC H3A 0C7, Canada. ✉email: martin.schmeing@mcgill.ca

Nonribosomal peptides are a large group of natural products, bioactive secondary metabolites that are often useful to society as therapeutics and green chemicals[1,2]. They include important medicines such as immunosuppressants (rapamycin), antivirals (cyclosporin A), antibiotics (daptomycin), antifungals (caspofungin), and antitumors (bleomycin)[3–5]. Despite their modest size of ~2–18 residues, nonribosomal peptides exhibit an impressive range of bioactivities because they can occupy a relatively large volume of chemical space. Nonribosomal peptides can contain many residues not typically found in proteins, such as D-, methylated, halogenated, and other nonproteogenic aminoacyl residues, aryl acyl residues, fatty acyl residues, and hydroxy acyl residues. Furthermore, nonribosomal peptides often contain heterocycles or are macrocyclic or branched. These topologies provide advantages such as pre-organization for binding targets or protease resistance[6–10]. The varied, powerful bioactivities and interesting chemical structures of nonribosomal peptides have led to a large number of studies on the total synthesis of the peptides and on biosynthesis and bioengineering of the enzymes which make them[11].

Nonribosomal peptides are made in microbes by elegant biosynthetic megaenzymes called nonribosomal peptide synthetases (NRPSs)[12,13]. NRPSs are organized as assembly lines of repeating sets of domains. Each set of domains, known as a module, is responsible for adding one acyl monomer substrate, typically an amino acid residue, to the growing peptide chain[1]. A minimal NRPS elongation module contains three domains: the adenylation (A) domain selectively binds the monomer substrate, activates it by adenylation, and transfers it as an aminoacyl thioester to the prosthetic phosphopantetheine (ppant) moiety on the peptidyl carrier protein (PCP) domain. The PCP domain transports the covalently bound amino acid to the condensation (C) domain. The C domain catalyzes peptide bond formation between the aminoacyl residue on this PCP and the peptidyl moiety on the upstream PCP$_{n-1}$ domain, elongating the peptide chain. The PCP domain, now with the newly elongated nascent peptide, translocates to the downstream module, where the peptide is further elongated and passed downstream in the next condensation reaction.

Variations and additions to basic NRPS biosynthesis contribute to the diversity of nonribosomal peptides. These variations can occur before, during, and after NRPS assembly-line synthesis. First, NRPS substrates can be produced from cellular metabolites by dedicated, pre-assembly-line synthetic enzymes[14,15]. Second, tailoring can occur during NRPS assembly-line synthesis, as an additional step in the catalytic cycle of a module. This can be encoded into the NRPS enzyme itself: the NRPS module can contain an alternative tailoring domain such as the heterocyclization (Cy) domain, which replaces the C domain and performs both condensation and heterocyclization[16–18], or the NRPS module can include an additional tailoring domain such as a reductase, methylation, monooxygenases, oxidase or formylation domain, which acts on a biosynthetic intermediate tethered to the PCP domain[19–21]. Tailoring during NRPS assembly-line synthesis can also be catalyzed by separately encoded enzymes: Enzymes including halogenases, oxidases, hydroxylase, and P450s[22–26] can interact non-covalently with NRPSs and act on PCP-tethered intermediates. Third, after the nonribosomal peptide is released from the NRPS, one or more separately encoded enzymes can catalyze additional reactions to yield the final natural product[27]. Tailoring after release from the NRPS occurs in chloroeremomycin, vancomycin, and penicillin biosynthesis[28–30].

A full biosynthetic pathway for a nonribosomal peptide can thus involve as few as one enzyme (an NRPS), or may require NRPSs as well as other enzymes for pre-, co-, and/or post-assembly-line synthetic steps. Genes for such enzymes are often found together with the NRPS genes in the producer microbe, in a biosynthetic gene cluster (BGC)[31]. However, from the sequence of a BGC it is difficult to predict whether a separately encoded enzyme will act on an acyl substrate before assembly-line synthesis, on a PCP-tethered intermediate of NRPS assembly-line synthesis, or on a small molecule after release from the NRPS. The natures of these possible substrates are quite different, and an enzyme will act on only one of these species[24,32].

The bacillamide biosynthetic pathway (Fig. 1a) is a three-protein BGC that contains genes for BmdA, tryptophan decarboxylase which converts L-Trp into tryptamine (Tpm)[33]; BmdB, an NRPS with domains A$_1$-PCP$_1$-Cy$_2$-A$_2$-PCP$_2$-C$_3$ that include alternative tailoring domain Cy$_2$, which installs a thiazoline ring in the peptide intermediate[16]; and BmdC, which had not been studied but can be recognized by its sequence as an oxidase enzyme in the nitro-FMN reductase superfamily[34,35]. Mature bacillamides have all been reported to contain thiazole rings[36–39], and thus BmdC appeared likely to oxidize the thiazoline ring to the thiazole seen in bacillamide D (Fig. 1a), but there is no indication of the timing of this oxidation.

Here, we describe studies on bacillamide synthesis that reveal unexpected biochemical aspects of the system, including stable dimerization induced by a tailoring protein, and an elegant NRPS-tailoring protein structure, previously undescribed for any nonribosomal peptide synthesis.

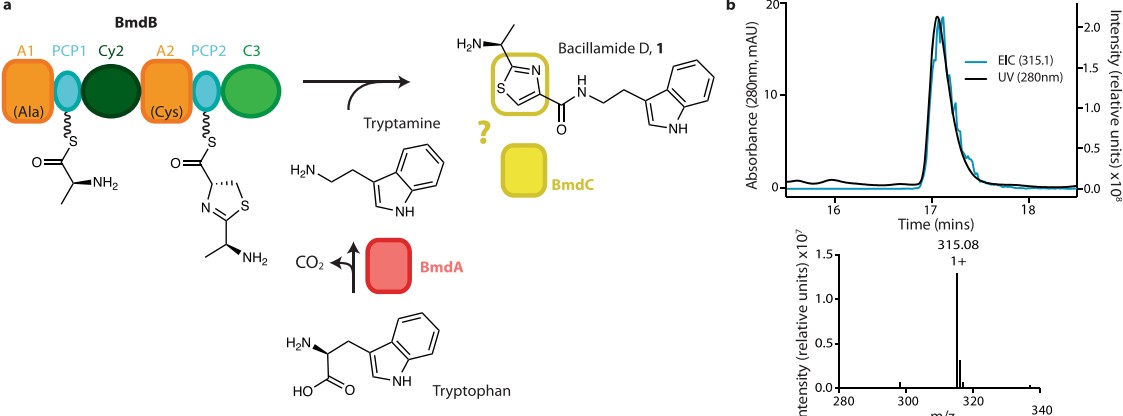

**Fig. 1 Biosynthesis of bacillamide D. a** A schematic drawing of the biosynthetic pathway of bacillamide D. The domains that make up BmdB are colored orange (adenylation domains), cyan (PCP domains), dark green (heterocyclization (Cy) domain), and green (terminal C domain). BmdC is depicted in yellow and BmdA in red. The color scheme is followed for all figures. **b** Mass spectrometry analysis of in vitro reconstitution of the biosynthesis of bacillamide D by BmdA, BmdB, and BmdC. Supplementary Fig. 1 shows the series of bacillamide biosynthesis reactions including controls.

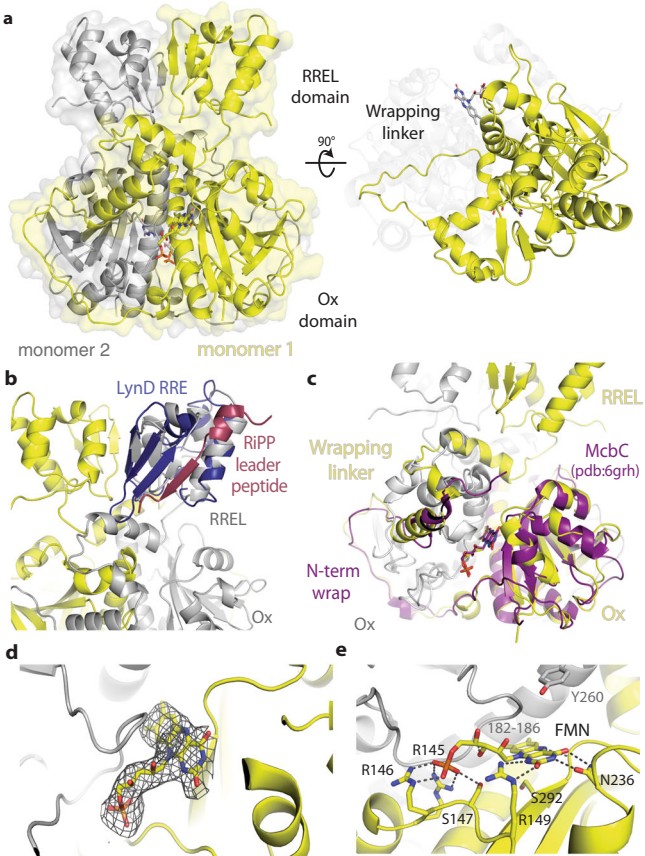

**Fig. 2 The structure of the oxidase enzyme BmdC. a** BmdC is a dimeric, di-domain enzyme. One protomer is shown in yellow and the symmetry mate that completes the biological dimer is shown in gray. **b** Superimposition of the RiPP recognition element (RRE) of LynD (blue; PDB code: 4v1t)[46] on the RREL domain of BmdC. The RiPP leader peptide bound to the LynD RRE is shown in red. **c** Superimposition of the Ox domain of McbC (purple, PDB: 6grh)[42] on the Ox domain of BmdC shows the structural similarities including the flavodoxin fold and extended wrapping linker. **d** Simulated annealing omit $F_O$-$F_C$ electron density map showing binding of FMN cofactor to BmdC. **e** The FMN binding site of BmdC.

## Results

**In vitro reconstitution of the bacillamide D biosynthetic pathway.** The bacillamide BGC is one of the best-represented pathways in genomic databases. Database searches return >400 clusters encoding putative proteins with high sequence identity to BmdA, BmdB, and BmdC, including ~350 clusters which show >65% protein identity. The clusters are mostly found in Bacilli, though other Bacillaceae are represented, such as *Laceyella* and *Thermoactinomyces* species. Yuwen et al. showed *B. atrophaeus* BmdA is a pyridoxal-5'-phosphate (PLP)-dependent tryptophan decarboxylase[33], and we showed *Thermoactinomyces vulgaris* F-5595 BmdB produces pro-bacillamide (AlaCys$_{thiazoline}$Tpm) in NRPS assembly-line synthesis[16] (Fig. 1).

To perform in vitro reconstitution of the full bacillamide D biosynthetic pathway, we heterologously expressed *T. vulgaris* BmdA, BmdB and BmdC in *E. coli* individually and purified them to homogeneity (Supplementary Fig. 1a). Reactions containing BmdA, BmdB and BmdC, PLP, ATP, alanine, cysteine, and tryptophan displayed robust production of a species of $[M + H^+] = 315.14$ (Fig. 1b). This mass and comparison to synthetic standard confirmed this compound as AlaCys$_{thiazole}$Tpm, i.e., bacillamide D (compound **1**) (Supplementary Fig. 1b), indicating that the full bacillamide D pathway was reconstituted in vitro. Reactions which contained

tryptamine in place of BmdA, PLP and tryptophan also showed robust bacillamide D production (Supplementary Fig. 1e), indicating that BmdA can fulfill its role simply by producing tryptamine, and that interaction of BmdA with BmdB or BmdC is not required. Removal of BmdC from the reaction conditions gave a species with $[M + H^+] = 317.15$, identified as AlaCys$_{thiazoline}$Tpm (pro-bacillamide, compound **2**)[16] (Supplementary Fig. 1f–h). These in vitro reconstitutions confirm BmdC as the oxidase involved in the bacillamide biosynthetic pathway (Fig. 1).

**The structure of the oxidase BmdC.** We performed X-ray crystallography of BmdC to gain a better understanding of its structure and function. Highly purified BmdC was subjected to sparse array crystallization. Initial yellow-colored crystals were obtained, consistent with the presence of a flavin cofactor. Iterative optimization gave crystals that yielded high-quality diffraction datasets (Supplementary Table 1). Molecular replacement using a nitro-FMN reductase domain from *Anabaena variabilis* (PDB ID 3eo7, unpublished) as a search model allowed structure determination at 2.7 Å resolution (Supplementary Table 1, PDB:7ly6). A single copy of BmdC is present in the asymmetric unit (Fig. 2 and Supplementary Fig. 2), and the adjacent symmetrically related copy completes the biological dimer. Nitro-FMN reductase domain protein typically exists as dimers or pseudodimers[40], and size-exclusion chromatography experiments reveal that BmdC is dimeric in solution (Supplementary Fig. 2e).

The structure of BmdC shows it to be a dimer of di-domain protomers (Fig. 2 and Supplementary Fig. 2). The small N-terminal domain contains 96 residues and features a winged helix–turn–helix motif (wHTH) with a very small β-sheet. A DALI[41] structural similarity search reveals this domain to be similar to the precursor peptide recognition elements (RRE) in ribosomally synthesized post-translationally modified peptide (RiPP) processing enzymes with Z-scores ranging from 5.3 to 6.5 and rmsd values of 2.0–2.9 Å over 61–69 alpha carbons (Fig. 2b)[42–46]. These RREs are often found in RiPP pathways for the production of linear azole-containing peptides, azole-containing cyanobactins and thiopeptides[43]. The larger C-terminal (Ox) domain is comprised of residues 142–325 and folds into a six-stranded β-sheet sandwiched between α-helices on both sides, similarly to the flavodoxin-like fold in other nitro-FMN reductases[47]. BmdC contains a 45-residue loop segment between the RRE-like (RREL) and Ox domains, that wraps around the Ox domain of the other protomer. The loop contributes 1764.3 Å$^2$ of the total 5777.3 Å$^2$ per-protomer surface area buried by BmdC dimerization[48,49]. These very high values of buried surface area are consistent with our observation that BmdC exists as a dimer in solution. The overall structure of BmdC is most similar to ThcOx (PDB: 5lq4), an FMN-dependent oxidase involved in the production of the cyanobactin patellamide[50], though ThcOx contains additional copies of the RRE and does not contain the wrapping loop (Supplementary Fig. 2c). McbC (PDB: 6grh), an oxidase involved in microcin B17 biosynthesis[42], does not contain an N-terminal domain, but does have a similarly dramatic wrapping loop (Fig. 2c).

The electron density maps showed very strong density for flavin mononucleotide (FMN) (Fig. 2d). Because the cofactor was co-purified with BmdC, we confirmed its identity through heat denaturation followed by mass spectrometry, which showed its $[M + H^+]$ of 457.1, characteristic of FMN and ruling out FAD (Supplementary Fig. 2d). As with other dimeric flavodoxins, BmdC binds FMN in an active site between its protomers (Fig. 2e). One protomer contributes to most of the side-chain interactions, with both the FMN phosphate, (through Arg145,

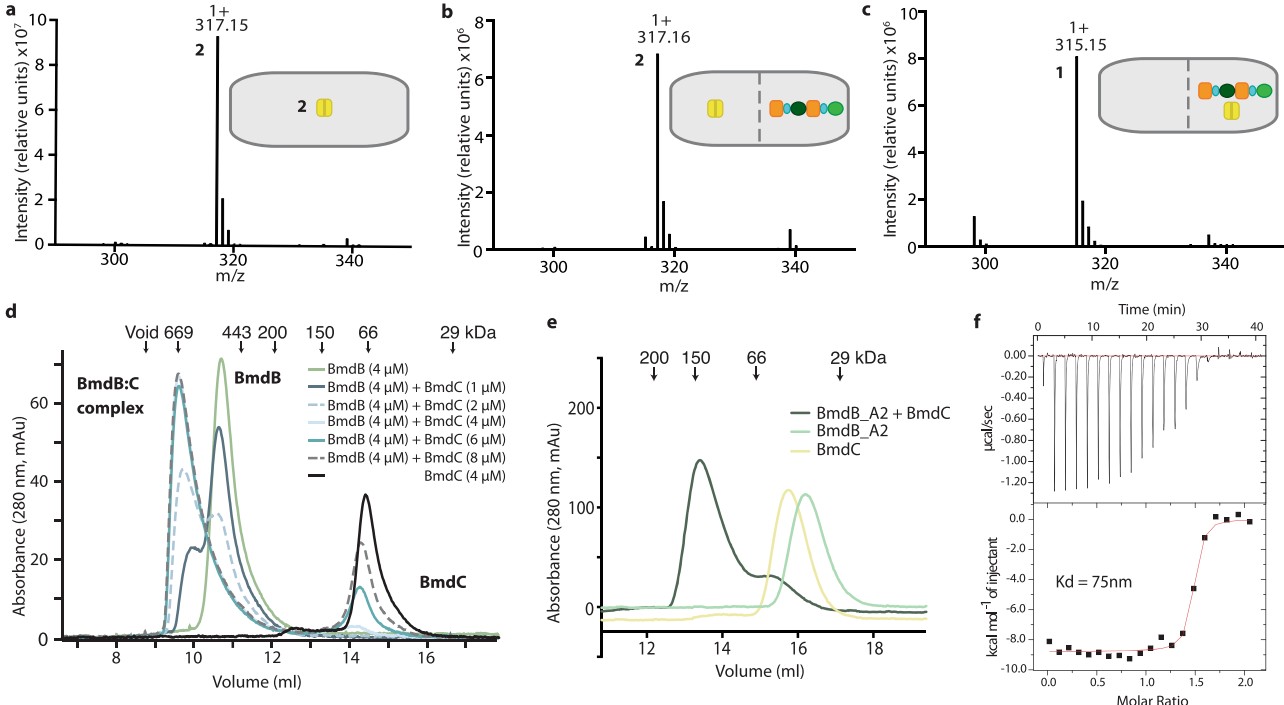

**Fig. 3 BmdC must interact with BmdB for oxidation and dimerizes BmdB. a** BmdC does not oxidize pro-bacillamide (**2**), presented as a free small molecule substrate. (Reactions in **a**, **b**, and **c** performed in triplicate with similar results.). **b**, **c** Dialysis experiments show that when BmdC and BmdB are (**b**) separated by a 10 kDa MW membrane, pro-bacillamide (**2**) is produced, while when (**c**) BmdB and BmdC are present on the same side of the membrane, the oxidized bacillamide D (**1**) is produced. **d** Size-exclusion experiments show that the addition of increasing amounts of BmdC lead to the formation of a BmdB:C complex with an elution volume corresponding to binding of two BmdB molecules to a BmdC dimer. **e** Size-exclusion experiments show BmdC binds to the $A_2$ domain of BmdB. **f** Isothermal calorimetry experiments where $A_2$ domain of BmdB is titrated by injection into a solution of dimeric BmdC show binding at nanomolar affinity. These experiments were performed in triplicate and with Kd values ranging from 44 nM to 118 nM.

Arg146 and Ser147) and the catalytic isoalloxazine rings (hydrogen bonding of Arg149 with FMN N1 and O2, Asn236 with N3 and O4, plus Gly294 with N5; analogous to interactions observed in flavin reductase P[51]). The second protomer contributes backbone interactions through residues 182-186. The isoalloxazine ring system appears planar, but ultra-high-resolution would be needed to definitively show planarity[51]. The conserved active-site Tyr260 points toward the FMN in each protomer. The analogous tyrosine in indigoidine synthetase has been proposed as a general base for oxidation, and BmdC is likely to use the same catalytic mechanism[52]. There is room for the thiazole ring to bind between this Tyr260 residue and the catalytic FMN N5 atom; a glycine from the crystallization buffer is observed occupying this site.

**BmdC acts during BmdB assembly-line synthesis and induces BmdB dimerization.** Inspection of the bacillamide BGC and of the BmdC structure does not clearly indicate whether oxidation by BmdC occurs during assembly-line synthesis or after small molecule release from the NRPS. A reaction mixture of 1 μM BmdC and 2 mM pro-bacillamide showed no production of bacillamide D (Fig. 3a), suggesting BmdC does not act on free pro-bacillamide. To test directly whether the oxidation occurs with pro-bacillamide or with AlaCys$_{thiazoline}$-$S$-BmdB as the substrate for BmdC, we designed dialysis experiments. Using dialysis membrane which allows small molecules but not proteins to pass through, we saw that bacillamide D production occurs only when BmdB and BmdC are found on the same side of the dialysis membrane, not when they are separated (Fig. 3a–c). When BmdB and BmdC are separated, only pro-bacillamide is detected. These results suggest that BmdC acts during NRPS

assembly-line synthesis and show that it must physically interact with BmdB for its function.

To investigate the interaction between the oxidase BmdC and the NRPS BmdB, we performed size-exclusion chromatography titration experiments. Mixing increasing amounts of BmdC with fixed amounts of BmdB led to a distinct peak shift corresponding to a BmdB-C complex (Fig. 3d). Interestingly, the BmdB-C complex elutes from the column substantially earlier than expected for a dimer of BmdC binding to a single NRPS. Rather, it corresponds to ~650 kDa, consistent with a complex comprised of two molecules of BmdB and a dimer of BmdC (605 kDa). This dimerization is unusual, as NRPS systems usually act as monomers[53].

Although dimerization in NRPSs is rare, polyketide (PKS) systems are typically dimers[54–57]. Complementation experiments, where each protomer of a dimer has a different inactivating mutation (for example in the conserved serines of the carrier proteins (CP) or the active-site cysteines in ketosynthase domains), have been used to show that the growing polyketide passed from one protomer to the other during synthesis[58–60]. We created mutants of BmdB that prevent pantetheinylation on PCP$_1$ (BmdB_S792A) and PCP$_2$ (BmdB_S1820A), respectively (Supplementary Fig. 3a, b). Neither BmdB_S792A nor BmdB_S1820A produced pro-bacillamide. Crucially, mixing BmdC with BmdB_S792A and BmdB_S1820A did not lead to substantial bacillamide production either, indicating that the two BmdB mutants could not complement each other in the context of a BmdC-induced dimer, and that it is unlikely the nascent peptide is passed between protomers.

We next compared rates of peptide synthesis of pro-bacillamide and bacillamide D by monomeric and dimeric BmdB(-BmdC). The reconstitution experiments described above show similar total production of pro-bacillamide by BmdB and

bacillamide D by BmdB–BmdC, but that BmdB–BmdC proceeds with around twice the initial rate (4.6 min$^{-1}$ vs 2.5 min$^{-1}$; Supplementary Fig. 3a–f). The rate difference is likely because of a preference for a thiazole intermediate over a thiazoline intermediate for condensation with tryptamine at the C3 domain of BmdB. To discern more directly whether dimerization and/or oxidation has impact on the overall rate of synthesis, we created an FMN-free BmdC. BmdC double mutant R146E S292F does not co-purify with FMN, but still induced dimerization of BmdB. Peptide synthesis reactions showed that BmdC_R146E-S292F-induced dimeric BmdB produces pro-bacillamide at the same rate as (monomeric) BmdB alone (Supplementary Fig. 3f). Therefore, dimerization does not appear to impart a catalytic advantage to bacillamide synthetase.

To identify the domain of BmdB to which BmdC binds, we made a series of BmdB truncation constructs and tested their ability to bind BmdC by gel filtration. This showed BmdC binds the A domain of the second module of BmdB (BmdB$_{M2}$) (Fig. 3e and Supplementary Fig. 4). We confirmed this interaction by isothermal titration calorimetry (Fig. 3f).

**Structural investigations of the BmdB$_{M2}$–BmdC complex.** The complex of BmdB$_{M2}$–BmdC was subjected to structural investigation by X-ray crystallography and cryo-electron microscopy. BmdB$_{M2}$ was expressed, purified, modified with various substrate analogues, bound with BmdC, and the complex purified by gel filtration prior to crystallization screening and/or grid preparation. We present three resulting structures: a BmdB$_{M2}$–BmdC complex structure with a cysteine-vinylsulfonamide adenylate[61,62], determined by X-ray crystallography at 3.8 Å resolution (PDB:7ly7), a BmdB$_{M2}$–BmdC complex structure determined by cryo-EM at 3.8 Å resolution (PDB:7ly4), and an in situ proteolyzed complex of two-thirds of the BmdB A$_2$ domain with the Ox domain of BmdC, determined by X-ray crystallography at 2.5 Å resolution (PDB:7ly5) (Fig. 4, Supplementary Fig. 5, Supplementary Fig. 6, and Supplementary Tables 1–3).

BmdB$_{M2}$–BmdC complex is a very elongated dimeric structure in which the two BmdB modules are kept far apart, and in which the active sites accessed by PCP$_2$ form a catalytic arc (Fig. 4). The center of the complex is BmdC in the same dimeric configuration described above. The area of the BmdC Ox domain most distal from the dimeric interface (residues 223–231 and 303–306) makes a small interface with one end of the A domain (residues 1321–1323, 1468, 1512–1513), burying 491 Å$^2$ of surface area per protein. This positions the two BmdB$_{M2}$ modules in a plane perpendicular to the BmdC rotational axis, stretching out at ~45° to the core BmdC dimer interface. The configuration is extremely oblong, with the BmdB$_{M2}$:BmdC complex having overall dimensions of 65 Å by 65 Å by 280 Å and placing the Cy$_2$ domains on either side of the massive complex. There is some flexibility in this configuration, with the relative Ox:A$_2$ orientation varying by ~11° across the three structures. The Cy$_2$:A$_2$ interface is largely constant and the distal N-lobe of Cy$_2$ substantially weaker in both EM and crystallography maps, indicative of both N-lobe:C-lobe flexibility[63,64] and overall variation in the periphery of the complex. Furthermore, though both BmdB modules of the BmdB$_{M2}$-BmdC complex are visible in EM maps, the highest quality map was obtained by performing refinement with a mask that enveloped the BmdC dimer and a single BmdB$_{M2}$ protomer, because the modest flexibility of the Ox:A$_2$ interface does break the overall twofold symmetry of the full complex. Notably, the cryo-EM map was calculated using a sample in which the PCP domain was not locked into a single conformation, and as a result, the PCP and A$_{sub}$ domains are not visible, despite attempts at focused classification around PCP-binding sites. The EM maps

do confirm the overall conformation and mode of dimerization observed by crystallography. In the BmdB$_{M2}$–BmdC crystal structure, the cysteine-vinylsulfonamide-adenylate biases the PCP toward the thiolation state[62,65], and PCP$_2$ is visible in this position (Fig. 4a) with the domain-domain interaction dominated by the A domain helix of residues 1520–1532 and the PCP helix of residues 1820–1833. Surprisingly, A$_{2sub}$ is disordered, despite no visible impediment to its binding in its classic thiolation position[66], suggesting A$_{2sub}$ may not be required to help position PCP$_2$ for aminoacylation.

In the catalytic cycle of module 2, PCP$_2$ must visit A$_2$ for thiolation/aminoacylation to ligate the cysteine substrate (forming Cys-S-PCP$_2$), then the Cy$_2$ domain for condensation and heterocyclization (forming AlaCys$_{thiazoline}$-S-PCP$_2$) and the BmdC Ox domain for oxidation (forming AlaCys$_{thiazole}$-S-PCP$_2$). The positions of the active sites are marked out by: the present BmdB-C crystal structure (step 1—thiolation), previous Cy$_2$ and LgrA PCP$_1$-C$_2$ containing structures[16,17,67] (steps 2 and 3—condensation and heterocyclization), and the position of FMN in BmdC (step 4—oxidation). These three active sites form an in-plane arc, with each opening facing the inside of the concave surface (Fig. 4c). This decreases the distance PCP$_2$ is required to travel to transport its intermediates.

The proteolyzed complex provides a high-resolution view of the core BmdB–BmdC interface (Fig. 4d). The BmdB–BmdC binding interface is quite small, burying 491 Å$^2$ of surface area per molecule in the full proteins. Of this, 423 Å$^2$ is present in the truncated complex[48,49]. Isothermal calorimetry binding experiments between A$_2$ and dimeric BmdC reveal a K$_d$ value of ~75 nM (Fig. 3f)[68]. The interaction features a network of hydrogen bonding, including two central salt bridges, between BmdC Asp225and BmdB Arg1513 and between BmdC Asp231 and BmdB Arg1512, plus side-chain-backbone and backbone-backbone contacts (Fig. 4c). Single point mutation of A$_2$(R1512D), A$_2$(R1513D), BmdC(D225R), and BmdC(D231R) each leads to disruption of the BmdB–BmdC complex, as observed by gel filtration (Fig. 4e). Peptide synthesis assays with BmdC(D225R) and wildtype BmdB decreased oxidation by ~50%, which confirms the importance of this BmdC–BmdB_A$_2$ interaction. It is not clear whether the remnant oxidation is because BmdC(D225R) retains some affinity to A$_2$ of BmdB (albeit not sufficient to maintain the complex through gel filtration), or is caused by the transient binding of PCP$_2$ to BmdC.

**Embedding the BmdC oxidase domain into BmdB.** NRPS-embedded Ox domains are thought to have arisen from the genetic insertion of genes for stand-alone Ox enzymes into NRPS genes. That the Ox domain of BmdC has homology with NRPS-embedded Ox domains[47,52,69] led us to ask whether we could replicate evolution and embed BmdC Ox into BmdB to turn it into an embedded tailoring domain. Searching the NCBI database of nonredundant proteins returned 40 proteins which include both RREL and Ox domains contained within a larger NRPS or PKS system. Thirteen showed similar insertion points within NPRS A domains and featured an average sequence identity of ~45% to BmdC. Sequence alignments with BmdB A$_2$ indicate that the embedded Ox domains are inserted at a point corresponding to that between A$_2$ loop residues Ser1534 and His1535 (Fig. 5). Notably, this loop is adjacent to the helix that includes BmdB Arg1512 and Arg1513 of the BmdB–BmdC interface. We, therefore, created fusion genes in which BmdC, flanked by linkers observed in embedded RREL-Ox didomains from *Paenibacillus tianmuensis* (Link*Pt*) or from *Bacillus pseudomycoides* (Link*Bp*), is inserted into full-length BmdB after residue 1534. By inserting the BmdC Ox domain without the RREL domain were we able

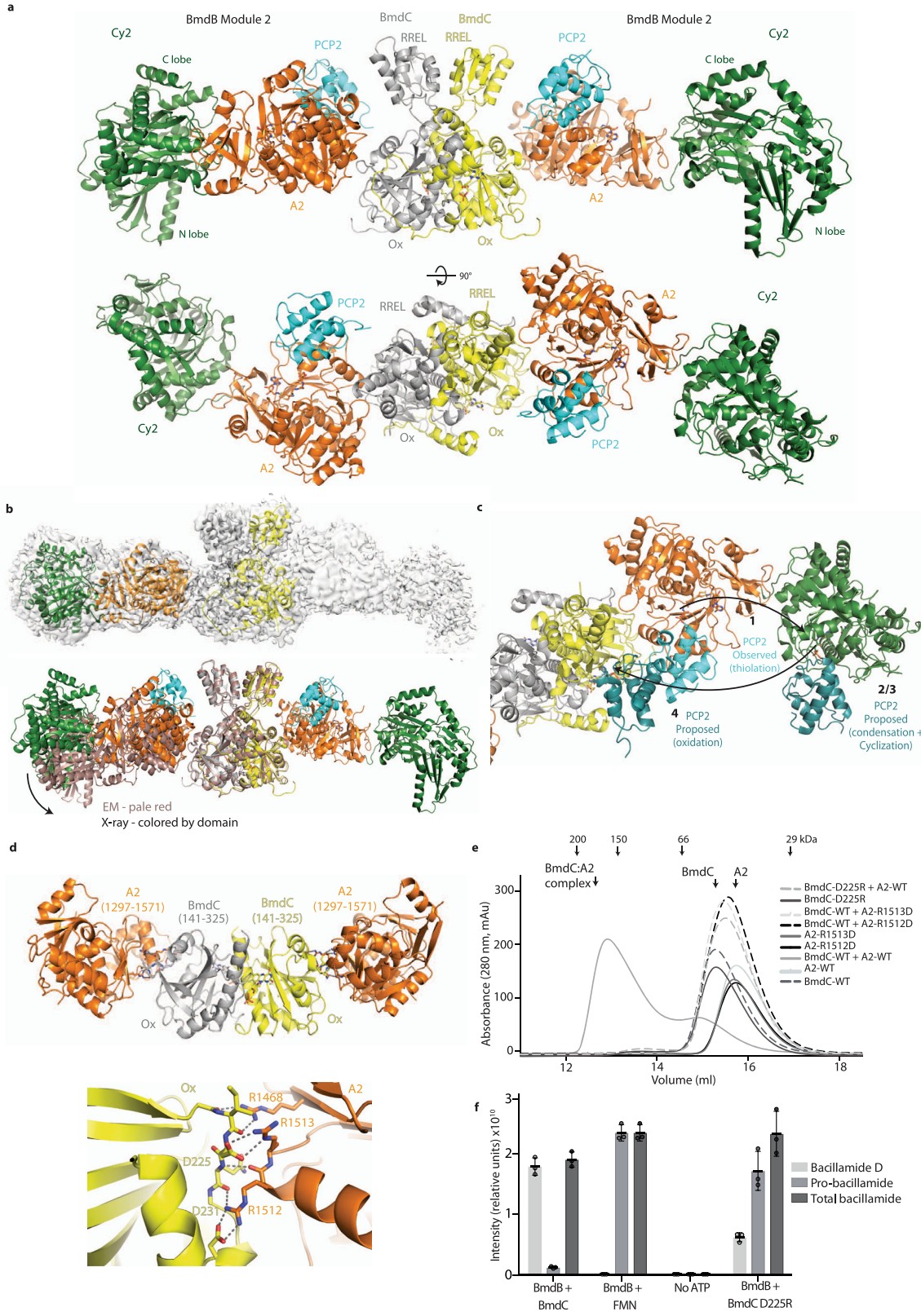

to achieve protein expression and purification. In vitro assays show that these constructs, BmdB–BmdC(Ox)Link*Bp* and BmdB–BmdC(Ox)Link*Pt*, are able to produce total bacillamide at 50% and 25% efficiency compared to native BmdB–BmdC. Although the total bacillamide produced by BmdB–BmdC(Ox) Link*Pt* is lower, it is more efficient at oxidation, with half the total bacillamide produced being pro-bacillamide and half bacillamide

D (Fig. 5). BmdB–BmdC(Ox)_Link*Bp* oxidizes ~30% of the bacillamide it produces.

## Discussion

Tailoring domains and proteins play an important part in allowing NRPS products to occupy such a large volume of

**Fig. 4 Co-complex structures of BmdB module 2 and BmdC. a** Overall structure of the biological dimer of BmdB$_{M2}$-BmdC solved by X-ray crystallography. The asymmetric unit contains one protomer each of BmdB and BmdC with symmetry mates completing the biological dimers. **b** Electron microscopy of the BmdB$_{M2}$-BmdC. Top: An EM map of the full dimeric BmdB$_{M2}$-BmdC (gray) was calculated at 4.2 Å and showed some variability in the relative positions of the BmdB$_{M2}$. Imposing a mask around the BmdC dimer and one copy of BmdB produced higher resolution maps, at 3.8 Å resolution, into which BmdC and BmdB Cy2A2 could be modeled. Bottom: Superimposition of BmdB$_{M2}$-BmdC structures determined by X-ray crystallography and by cryo-EM show some modest differences in the BmdC:BmdB interaction angle. **c** Proposed positions BmdB PCP2 assumes around the catalytic arc during the synthetic cycle in module 2. The position of PCP2 for thiolation (aminoacylation) (1) is observed in this study, the position for condensation and heterocyclization (2/3) is based on the condensation state of LgrA (PDB:6mfz)[67], and the position for oxidation is modeled based on the location of FMN in BmdC. **d** The structure of the in situ proteolyzed complex of BmdB_A$_2$ and BmdC Ox (top), and a close-up view of the A$_2$:Ox interface (bottom). **e** Mutations that disrupt either salt bridge in the BmdB:BmdC interface abolished binding between proteins in size-exclusion experiments. Complementing mutations are unable to rescue this interaction. **f** Oxidation to bacillamide D is still possible despite the loss in binding between BmdB and BmdC. The central value for the reactions represents the mean, while the standard deviation of the mean is represented by the error bars. Individual points of the triplicates are shown.

chemical space[28,70]. These domains and proteins are presumably evolutionarily co-opted to function with the NRPS as their genes have been physically taken into the BGC or spliced into the NRPS gene itself. The presence of a RiPP precursor peptide recognition-like element (RRE) as the N-terminal domain of BmdC raises the possibility that BmdC was co-opted from a RiPP cluster into the bacillamide BGC. The RRE binds its RiPP precursor peptide as a beta-strand along the edge of its β-sheet. This β-sheet is present in the RRE-like domain of BmdC, and overlaying RRE:peptide complexes shows this beta-strand would point the peptide towards the FMN at the BmdC active site (Fig. 2 and Supplementary Fig. 2b). The heterocyclic peptidyl substrate (Ala-Cys$_{thiazoline}$) contains no leader and is far too short to bind both the RRE-like strand and the active site. In addition, the PCP contains no β strands, suggesting this RRE feature is likely an interesting evolutionary relic left over from coopting an RiPP oxidase, rather than a functional element in bacillamide synthesis.

The biochemical (Figs. 3 and 4) and structural (Fig. 4) results suggest that oxidation during bacillamide synthesis requires substrate delivery to the Ox domain by PCP$_2$. It is likely that Ox:PCP$_2$ protein–protein interactions are required for productive substrate binding to the Ox active site. The BmdC structures show the active site of BmdC to be fairly wide and shallow, hinting that a small molecule such as pro-bacillamide may not be able to make extensive binding interactions.

Bacillamide synthetase BmdB exists as a BmdC-induced dimer. As mentioned, dimerization is an uncommon feature in exclusively NRPS systems[53,65], but is common in polyketide and fatty acid synthases[57,58,71]. One known example of a dimeric NRPS is in the vibriobactin biosynthesis pathway, where NRPS subunit VibF is dimerized through a catalytically inactive C domain[72]. Similar to the situation with BmdB, this dimerization is not necessary for function and does not significantly increase product formation[72]. Very recently, an elegant structure of a Cy-A-PCP construct of FmoA3 showed it has a head-to-tail homodimer architecture with a massive dimerization interface along the back side of the Cy and A domains[73]. The Cy-A domain interface observed in BmdB is similar to that seen in FmoA3[73] and in other C-A-containing structures[67,74–76]. The BmdB$_{M2}$–BmdC complex structures (Fig. 4) clearly show why dimerization through BmdC is neutral for bacillamide biosynthesis and why BmdB mutants do not complement each other: The BmdB–BmdC binding interaction holds the BmdB protomers very far apart, such that they are unlikely to interfere with each other. We suggest that dimerization of BmdB through BmdC is a product of the dimeric nature of BmdC, which is common in oxidases[40,51,77,78]. The twofold symmetry of BmdC presents two BmdB binding sites that can be occupied simultaneously, and because dimerization does not impart a catalytic disadvantage, there is no evolutionary pressure to eliminate it.

The relatively stable interface between BmdB A$_2$ and BmdC means that the flexibility of the BmdB$_{M2}$–BmdC interaction is

modest. There is clearly some flexibility visible in the EM dataset, so it is more straightforward to refine the region of the map consisting of a single BmdB$_{M2}$ and a dimer of BmdC, but the BmdB$_{M2}$ are in the same general orientation relative to BmdC. This feature of the dimerized module contrasts markedly with the dimodular NRPS linear gramicidin synthetase subunit A, where the first and second modules assume many different relative orientations[67]. The difference is explained by the fact that the two BmdB$_{M2}$ modules are bound to each other through a fairly consistent interaction with the Ox domain of BmdC, whereas modules 1 and 2 of LgrA are tethered by a very flexible linker.

Many separately encoded tailoring enzymes that act during assembly-line synthesis have been described and characterized biochemically[23–25]. However, only systems with P450 tailoring[29,79] and prolyl oxidation[80] have been structurally characterized, and they feature transient NRPS: tailoring enzymes interactions, rather than the stable tailoring complex reported here. Indeed, although Bmp3[80] and BmdC are both flavin-dependent oxidases of PCP-bound substrates, little other commonality is observed: Bmp3 has a completely different fold from BmdC, uses FAD, not FMN, is tetrameric, and needs to interact only with the type II PCP Bmp1, not with any other NRPS component. The two studies show similar oxidation function can be achieved in very different ways by different NRPS systems.

Oxidation during NRPS assembly-line synthesis can be catalyzed by separately encoded enzymes (in a stable complex like BmdB–BmdC, or a transient complex as in Bmp1:3) and by embedded domains. In addition to the RREL-Ox domains inserted into modules described above, famous examples of thiazole oxidation by embedded domains include those in the biosynthesis of bleomycin, epothilone, and indigoidine[69,81–86]. Bleomycin and epothilone are made by hybrid NRPS-PKS systems, so dimerization of the synthetases is not unexpected. The Ox domains are inserted within the A domain of EpoB and after the PCP domain of BlmIII respectively, and likely contribute to dimerization. As in EpoB, the Ox domain of indigoidine synthetase is inserted between A domain motifs A8 and A9, a common position for tailoring domains: The embedded methyltransferase in TioS is in this position and makes an apparently rigid interface at its insertion site within the A$_{sub}$ subdomain[87]. Interestingly, the inserted Ox sequence in indigoidine synthetase (IndC) is larger than that in EpoB (Supplementary Fig. 7b). Careful analysis and structure prediction using the Robetta web server[88] reveal the insertion is an Ox pseudodimer, suggesting IndC would not dimerize through the inserted pseudodimeric Ox domain. Indeed, gel filtration of purified IndC shows it is a monomer (Fig. 5). We suggest all indigoidine synthetases will have a pseudodimeric Ox which greatly resembles a BmdC dimer fused to their A domain (Supplementary Fig. 7c).

The in-plane catalytic arc of BmdB$_{M2}$:C (Fig. 4c) is reminiscent of catalytic chambers in polyketide and fatty acid synthases[89,90].

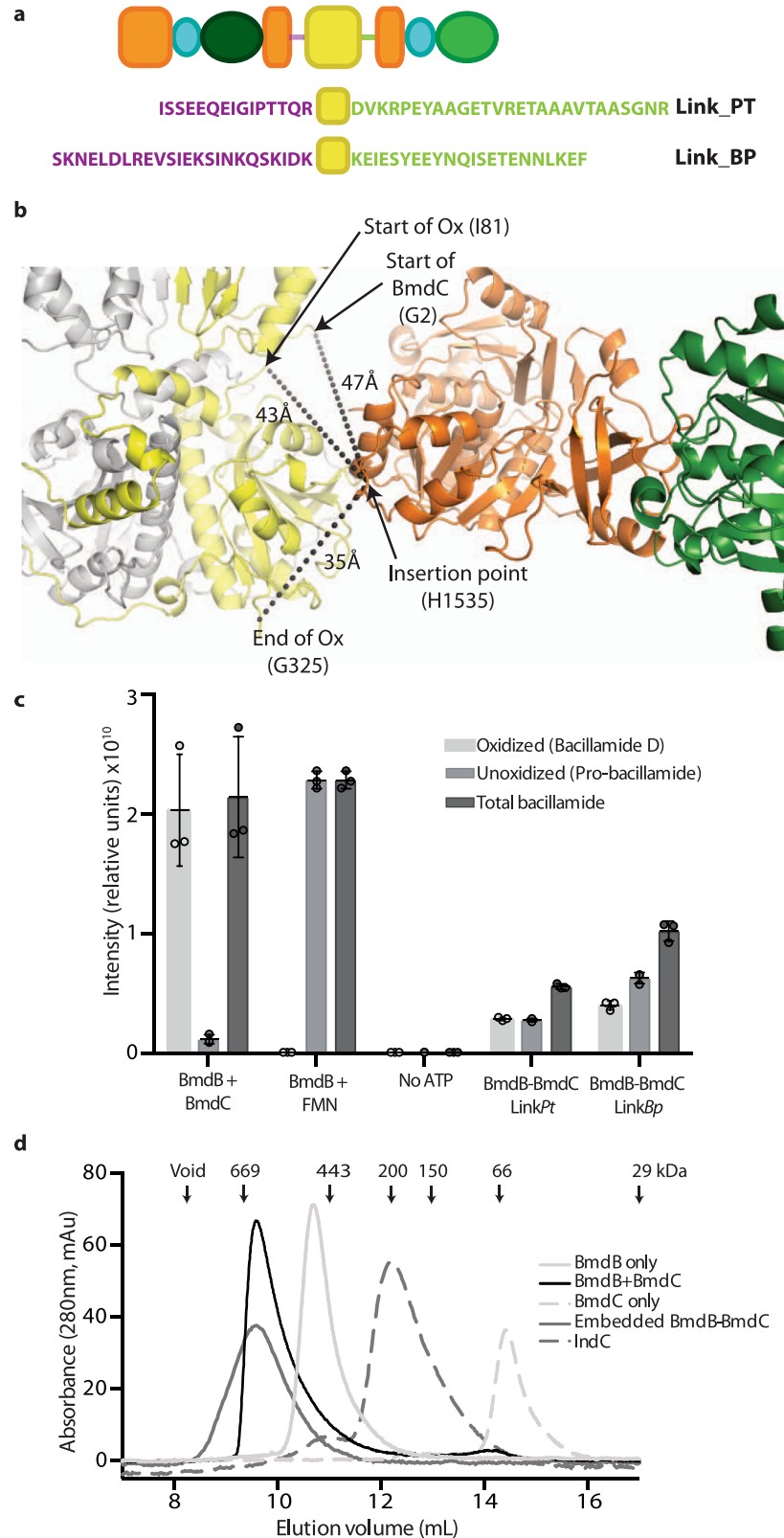

The arc seems to facilitate elegantly the transport function of PCP through the synthetic cycle of module 2 (Fig. 4c), while the BmdB:C geometry keeps each BmdB protomer separate. We believe the BmdB-C complex described here is the only nontransient tailoring complex characterized to date. It remains to be seen whether all such tailoring complexes, which provide a myriad of chemical modifications, have evolved equally elegant architectural solutions.

## Methods

**General reagents.** All commercial reagents purchased for this study are listed in Supplementary Table 4 and were used without additional purifications.

**Cloning and purification of BmdA, B, and C constructs.** The proteins used in this study are *Thermoactinomyces vulgaris F-5595* BmdA (WP_022737640.1; https://www.ncbi.nlm.nih.gov/protein/WP_022737640.1?report=genpept), BmdB

**Fig. 5 Embedded Ox domains. a** To insert BmdC or the Ox domain of BmdC into BmdB, the linkers with sequences of those found in NRPS with naturally embedded RREL-Ox didomains from *Paenibacillus tianmuensis* (Link_PT) or from *Bacillus pseudomycoides* (Link_BP) were used. **b** The insertion point of naturally occurring embedded RREL-Ox didomains corresponds to residue H1535 in BmdB. The distances to the beginning of the RREL domain, the beginning of the Ox and the end of the Ox domain in BmdC are shown. The number of residues in the chosen linkers is sufficient to span these distances if the linkers are in an extended conformation. **c** In vitro peptide synthesis experiments showing the production of bacillamide D and pro-bacillamide in wildtype constructs and the engineered embedded constructs ($n = 3$ independent reactions). The negative control reaction lacks ATP. The central value for the reactions represents the mean. while the standard deviation of the mean is represented by the error bars. **d** Gel-filtration experiments show that engineered embedded BmdB–BmdC constructs are dimeric, while IndC is monomeric in solution.

(WP_022737639.1; https://www.ncbi.nlm.nih.gov/protein/WP_022737639.1?report=genpept)[16] and BmdC (WP_022737638.1; https://www.ncbi.nlm.nih.gov/protein/WP_022737638.1?report=genpept), cloned from genomic DNA obtained from the Agricultural Research Service Culture Collection. Constructs were cloned into a pET21-derived vector containing an N-terminal TEV-cleavable calmodulin-binding peptide tag and a C-terminal TEV-cleavable octa-histidine tag (pBacTRev), or into a pET21-derived vector containing an N-terminal TEV-cleavable octa-histidine tag and a C-terminal TEV-cleavable calmodulin-binding peptide tag (pBacT). All PCR primer sequences, sources of amplified DNA, parental plasmids, restriction sites used, and resulting plasmid names for cloning in this study are listed in Supplementary Table 5. For all site-directed mutagenesis steps, plasmids and primer sequences are listed in Supplementary Table 6.

BmdA was expressed in BL21(DE3) *E. coli* in LB media containing 40 µg/ml kanamycin at 37 °C. When an $OD_{600}$ of ~0.5–0.6 was reached, the cultures were induced with 1 mM of IPTG and further grown at 16 °C overnight before harvesting by centrifugation. Cell pellets were resuspended in IMAC buffer A (50 mM Tris-HCl pH 7.5, 200 mM NaCl, 10 mM imidazole pH 8.0, 2 mM βME) and lysed by sonication. The lysate was clarified by centrifugation at 40,000×g for 20 min, the clarified lysate applied to a 1 ml HiTrap IMAC FF column charged with $Ni^{2+}$, and protein eluted with IMAC buffer B (50 mM Tris-HCl pH 7.5, 200 mM NaCl, 250 mM imidazole pH 8.0, 2 mM $CaCl_2$, 2 mM βME). Fractions containing BmdA were pooled and applied to a 20 ml calmodulin-binding peptide (CBP) column with equilibrated in CBP buffer A (50 mM Tris-HCl pH 7.5, 200 mM NaCl, 2 mM $CaCl_2$, 2 mM βME). BmdA was eluted with CBP buffer B (50 mM Tris-HCl pH 7.5, 200 mM NaCl, 2 mM EGTA pH 8.0, 2 mM βME), concentrated and applied to a Superdex200 10/300 column equilibrated GF buffer A plus 1 mM DTT.

BmdC was heterologously expressed with growth, harvest, and lysis as described for BmdA. Clarified lysate was applied to a 5 ml HiTrap IMAC FF column charged with $Ni^{2+}$, and BmdC eluted with IMAC buffer B. Fractions containing BmdC were pooled and applied to a 20-ml CBP column equilibrated with CBP buffer A, and eluted with CBP buffer B. Pooled fractions were dialyzed overnight against GF buffer A plus 2 mM βME, at which time affinity tags were cleaved with N-His-TEV protease (1 mg per 20 mg of BmdC) at 4 °C. The cleaved BmdC sample was passed again through the HiTrap IMAC FF and CBP columns, the flowthrough was concentrated and applied to a Superdex200 16/60 column equilibrated with GF buffer A (50 mM Tris-HCl pH 7.5, 200 mM NaCl) plus 2 mM βME.

BmdB was expressed and purified similarly to described[16]: protein was expressed in BL21(Bap1) *E. coli*[91], grown in LB media containing 40 µg/ml kanamycin at 37 °C before being induced, harvested, and purified in the same manner as BmdC.

BmdC_R146E-S292F, BmdC_D225R, BmdC_D231R, and BmdB_C3 were all expressed and purified as described for BmdC. BmdB_M2, BmdB_S792A, and BmdB_S1820A were expressed and purified by the same protocol described for BmdC except in BL21(Bap1) *E. coli*[91] and TEV digestion and re-application to affinity columns was not performed. BmdB_A2, BmdB_A2-R1512D, BmdB_A2-R1513D and BmdB_PCP2 were expressed and purified in as described for BmdC, but omitting CBP chromatography.

**Bacillamide synthesis assays.** Bacillamide synthesis reaction conditions were adapted from Duerfahrt et al.[92]. Reaction containing 1 µM BmdA, 1 µM BmdB, 1 µM BmdC, 1 mM L-alanine, 1 mM L-cysteine, 2 mM ATP, 1 mM pyridoxal-5'-phosphate (PLP), 2 mM FMN, 1 mM L-tryptophan, 50 mM HEPES pH 7.5, 100 mM NaCl, 0.5 mM TCEP and 10 mM $MgCl_2$ in a final volume of 50 µL were incubated for 2 h at 37ºC. Accompanying reactions lack components/substitute components as indicated in Supplementary Fig. 1 (e.g., no ATP controls; 2 mM tryptamine in place of PLP, tryptophan, and BmdA; no BmdC and FMN reaction). Assays depicted in Supplementary Fig. 1i were performed with 2 mM of pro-bacillamide, 2 mM ATP, 1 µM BmdC, 50 mM HEPES pH 7.5, 100 mM NaCl, 0.5 mM TCEP, and 10 mM $MgCl_2$ in the same volume. Reactions were stopped with 50 µL of 4:1 n-butanol:chloroform, lyophilized to dryness and resuspended in 50 µL of 10% methanol. Five microliters of sample were applied to a ZORBAX Extend-C18 (Agilent) column, analyzed with LC-ESI-MS and eluted with LC-method 1 (Supplementary Table 7). ESI-MS was performed with an in-line Bruker amaZon speed ETD ion-trap mass spectrometer (Fig. 1 and Supplementary Fig. 1). The pro-bacillamide and bacillamide D standards were purchased from Zamboni Chem Solutions. In vitro bacillamide reactions including BmdC_D225R were

performed the same way (Fig. 4f). When sets of experiments included quantification of bacillamide D and pro-bacillamide production (Figs. 4f and 5c), production was quantified using areas under the curves of extracted ion chromatograms (EICs). Bacillamide D was quantified using the area under the EIC for $m/z = 315.1$ across the time in which bacillamide peaks elute. Because some samples did not have full separation between peaks for bacillamide D and pro-bacillamide, quantification of pro-bacillamide was performed by integrating the area under the EIC for $m/z = 317.1$ and subtracting 6% of the area under the EIC for $m/z = 315.1$, to compensate for the $m/z = 317.1$ contribution from isotopic distribution of heavier isotopes within bacillamide D.

**BmdC crystallography.** Crystals of BmdC formed with the vapor diffusion, sitting-well technique at 22 °C with a protein concentration of 5 mg/ml and a crystallization solution of 35% PEG1500 (wt/vol), 0.1 M SPG buffer (0.148% (w/vol) succinic acid, 0.604% (w/vol) sodium dihydrogen phosphate monohydrate, 0.328% (w/vol) glycine, pH 9.0). Crystals were cryoprotected by the addition of 10% ethylene glycol, then looped and flash cooled in liquid nitrogen before data collection. Diffraction data were collected at the CLS 08-ID beamline of the CMCF at the Canadian Light Source ($\lambda = 0.979$ Å), in Saskatoon, Canada. The data were indexed in the space group $I2_13$ with iMosflm[93] and scaled with AIMLESS in CCP4[94]. Initial phases were obtained by molecular replacement using the program MORDA in CCP4[95] and search models homologous to the Ox domain, including 3EO7 (unpublished). The RREL portion was modeled using AutoBuild in PHENIX[96]. Geometry restraints for FMN were obtained with SKETCHER in CCP4[95]. The overall structure was then iteratively rebuilt in COOT[97] and refined in PHENIX[96] to produce the final structure (Supplementary Table 1). There is one molecule in the asymmetric unit, and a symmetry mate forms the biological dimer of BmdC.

**Cofactor identification.** BmdC was heat-denatured at 95 °C for 15 min and pelleted by centrifugation at 21,100×g (15,000 RPM with FA-45-24-11 rotor). The supernatant was filtered through a 10000 Da Amicon filtration device and the flowthrough used for HPLC[98]: Samples were applied on the ZORBAX Extend-C18 column and eluted with LC-method 2 (Supplementary Table 7). The cofactor eluted at ~10 min and was analyzed by direct injection and ESI-MS with an amaZon speed ETD ion-trap mass spectrometer. The standards of FAD and FMN used were purchased from Sigma-Aldrich.

**Dialysis experiments.** For dialysis experiments, 1 µM BmdB and 1 µM BmdC were placed on either the same side or different sides of dialysis membrane in a Slide-A-Lyzer (10,000 MWCO; Thermo Scientific), in reactions of 400 µL of 50 mM Tris-HCl pH 7.5, 100 mM NaCl, 10 mM $MgCl_2$, 2 mM FMN, 1 mM L-alanine, 1 mM L-cysteine, 10 mM tryptamine, and 2 mM ATP. The reactions proceeded for 2 hrs at 37 °C with 70 RPM of shaking, and were stopped with 200 µL of 4:1 n-butanol:chloroform. Samples were then lyophilized overnight before resuspension in 100 µL of 10% methanol. Samples (20 µL) were applied to a ZORBAX Extend-C18 (Agilent) column and eluted with LC-method 3 (Supplementary Table 7). ESI-MS was performed with a Bruker amaZon speed ETD ion-trap mass spectrometer.

**Gel-filtration binding experiments.** Full-length BmdB and BmdC at concentration of 1–8 µM were mixed and applied to a Superdex200 Increase 10/300 column equilibrated with GF buffer A plus 1 mM DTT (Fig. 3e). To identify the domain of BmdB which binds BmdC (Supplementary Fig. 4), 4 µM BmdB_M2, BmdB_A2PCP2, BmdB_Cy2[16], BmdB_A2, BmdB_PCP2, or BmdB_C3 were mixed with 4 µM of BmdC and applied to a Superdex 200 Increase 10/300 column equilibrated with GF buffer A plus 0.5 mM TCEP. For site-directed mutagenesis validation of the BmdB–BmdC binding interface by gel filtration (Fig. 4e), 5 µM of BmdB_A2, BmdB_A2-R1512D, or BmdB_A2-R1513D was incubated with 5 µM of BmdC, BmdC-D225R or BmdC-D231R and applied to the Superdex200 Increase 10/300 column equilibrated with GF buffer A plus 1 mM DTT.

**Bacillamide complementation experiments.** Complementation experiment reactions included 50 mM HEPES pH 7.5, 100 mM NaCl, 2 mM DTT, 10 mM $MgCl_2$, 1 mM L-alanine, 1 mM L-cysteine, 2 mM ATP, 10 mM tryptamine, 2 mM

FMN, 1 µM enzyme in a volume of 200 µL. Reactions were stopped with 400 µL of 4:1 n-butanol:chloroform before lyophilizing overnight and resuspending in 100 µL of 10% methanol for LC-MS analysis. Ten microlitres of the sample was applied to a ZORBAX Extend-C18 (Agilent) column and analyzed with LC-ESI-MS using LC-method 3.

**Assays with flavin-free BmdC.** Each reaction of 500 µL had 50 µL samples removed at 0, 1, 2, 5, 10, 15, 30, 60, and 120 min. The reaction contained 50 mM HEPES pH 7.5, 10 mM MgCl$_2$, 100 mM NaCl, 2 mM DTT, 1 µM enzyme(s), 1 mM L-alanine, 1 mM L-cysteine, 2 mM ATP, 10 mM tryptamine, plus 2 mM of FMN for the wildtype BmdC and BmdB reaction. FMN was not added to the BmdC_R146E_S292F because this mutant retains some affinity to FMN, but the 2 mM of FMN was confirmed not to promote oxidation (Supplementary Fig. 1). Reactions were stopped with 50 µL 4:1 n-butanol:chloroform before lyophilizing overnight and resuspending in 50 µL of 10% methanol before LC-MS analysis. The sample was centrifuged at 21,100×g (15,000 RPM with FA-45-24-11 rotor) before applying 10 µL of the supernatant to a ZORBAX Extend-C18 (Agilent) column and analyzing with LC-method 3.

Initial rates were estimated by performing a linear regression analysis (with GraphPad Prism version 6.0.0) from 0–30 min intersecting at $y = 0$, and converting from µM/min to moles of bacillamide produced per mole of bacillamide synthetase per minute. The apparent initial rate of bacillamide syntheses are BmdB: 2.5 min$^{-1}$; BmdB–BmdC: 4.6 min$^{-1}$; BmdB-BmdC_R146E_S292F: 2.3 min$^{-1}$.

**ITC experiments.** Samples were dialyzed overnight in a 1 L solution of GF buffer A plus 1 mM TCEP at 4 °C. The protein concentrations were measured using the NanoDrop 2000 spectrophotometer (Thermo Scientific) and extinction coefficients of 87140 M$^{-1}$ cm$^{-1}$ for the dimeric BmdC and 69010 M$^{-1}$ cm$^{-1}$ for BmdB_A$_2$. ITC was performed with a MicroCal iTC200 (GE Healthcare) with a stirring speed of 750 RPM, 5 µcals of reference power applied to the reference cell and 2 µL injections interspaced by 120 s in 200 µL of sample at 25 °C. The first injection of 0.4 µL was removed from the downstream analysis with the software package Microcal Origin 7.0 (OriginLab, Northhampton, MA). Analysis of the titration curves of these experiments was performed with a binding model using a stoichiometry of 1. The ITC experiment between wildtype BmdC and BmdB A$_2$ was performed by injecting 600 µM of BmdB A$_2$ into 60 µM of dimeric BmdC. These experiments were done in triplicate.

**Crystallization of the proteolyzed BmdB–BmdC complex.** The proteolyzed BmdB–BmdC complex is a result of unexplained proteolysis of both BmdB-A$_2$PCP$_2$ and BmdC during crystallization. BmdB_A$_2$PCP$_2$ was expressed and purified as described for BmdB_A$_2$, but adding an anion exchange step before gel filtration, using a Mono Q HR 16/10 column equilibrated in a Q buffer A and a gradient to Q buffer B over 100 mL. The eluted sample was concentrated and applied to a Superdex200 column equilibrated with GF buffer A with 1 mM TCEP.

BmdB_A$_2$PCP$_2$ was post-translationally modified using AlaCys$_{thiazole}$-amino-CoA, converted[99] from AlaCys$_{thiazole}$-amino-ppant (kindly gifted by Jon Patteson and Dr. Bo Li—University of North Carolina Chapel Hill), but the proteolysis removed the PCP$_2$ domain from the crystallized complex. A$_2$PCP$_2$ was incubated with a twofold molar excess of BmdC and applied to Superdex200 Increase 10/300 column with GF buffer B (50 mM Tris-HCl pH 7.5, 150 mM NaCl) with 1 mM TCEP. Crystals were formed by mixing 4.5 mg/ml of complex with a crystallization solution of 50 mM Tris-HCl pH 7.5, 160 mM KCl, and 21% PEG3350 (wt/vol), using sitting-drop vapor diffusion at 22 °C. The crystals appeared after ~12 days and only appear in 24-well sitting drops with drop ratios of 2 µL of protein and 1 µL of mother liquor. Crystals were cryoprotected by dipping the crystal in a solution containing 160 mM KCl, 21% PEG3350 (wt/vol) and 20% ethylene glycol before flash vitrification in liquid nitrogen. Diffraction data were collected using the 24-ID-E beamline of the NE-CAT at the Advanced Photon Source (APS) in Argonne, Illinois. The data was indexed and scaled with HKL2000[100] in space group H32. The phases were obtained by molecular replacement in PHENIX[96] using one chain of BmdC and a homology model of the BmdB A$_2$ domain generated by SWISS-MODEL[101] as search models. The overall structure was then iteratively rebuilt with COOT[97] and refined with PHENIX[96]. There is one molecule in the asymmetric unit and a symmetry mate forms the biological dimer of the proteolyzed complex of BmdB and BmdC.

**Crystallization experiments with BmdB$_{M2}$–BmdC complex.** BmdB$_{M2}$ was heterologously expressed for structure determination from pBacT_BmdB_M2_struct in BL21(EntD-) E. coli[102] in LB-kan media. BmdB$_{M2}$ was purified as described for BmdC, but adding an anion exchange step before gel filtration, using a Mono Q HR 16/10 column equilibrated in a Q buffer A (50 mM Tris-HCl pH 7.5, 2 mM βME) and a gradient to Q buffer B (a Q buffer A plus 1 M NaCl) over 150 mL. The eluted sample was concentrated and applied to a Superdex200 column equilibrated with GF buffer A with 1 mM TCEP.

The purified BmdB$_{M2}$ protein was first modified with ppant from coenzyme A (BioShop Canada Inc.) with 1 µM SFP, 10 mM MgCl$_2$, and a twofold molar excess of coenzyme A for 1 h. To remove the excess SFP and coenzyme A, the sample was applied to a Superdex 200 Increase 10/300 column equilibrated with GF buffer B

with 1 mM TCEP. Cysteine-vinylsulfonamide adenylate inhibitor (Zamboni Chem Solutions) was used to lock PCP$_2$ in the thiolation state in a reaction with 10 mM MgCl$_2$ and a threefold molar excess of the inhibitor at room temperature overnight. This reaction was then applied to the Superdex 200 Increase 10/300 column equilibrated with the same buffer to separate the excess inhibitor. BmdB$_{M2}$ was incubated with twofold molar excess of purified BmdC and applied to the Superdex 200 Increase 10/300 column equilibrated with GF buffer B with 1 mM TCEP. BmdB$_{M2}$:BmdC was crystallized by sitting-drop vapor diffusion after 7 days at 22 °C using 2 µL of 5.6 mg/ml protein complex and 1 µL of crystallization solution (0.2 M sodium citrate, 14% PEG3350, 5.6 mg/ml, 0.08 M guanidine hydrochloride and 0.1 M Bis Tris propane pH 6.1). Cryoprotection was accomplished by increasing the amount of MPD, using 2% steps, to a final concentration of 10%, before flash cooling in liquid nitrogen. Diffraction data from this crystal was collected at the CLS 08-ID beamline of the CMCF at the Canadian Light Source ($\lambda = 0.979$ Å), in Saskatoon, Canada. The data was indexed with iMosflm[93] to P3 and scaled with AIMLESS to P6$_5$22 in CCP4[94]. Initial phases were obtained by molecular replacement in PHENIX[96] using one copy of BmdC, one copy of the A$_2$ domain fragment solved of the proteolyzed complex structure and one copy of BmdB_Cy$_2$ (PDB: 5t3e) as search models. Geometry restraints for the cysteine-vinylsulfonamide adenylate inhibitor were generated with a the combined used of SKETCHER in CCP4[94] and REEL in PHENIX[96]. A homology model of the PCP$_2$ domain was generated using SWISS-MODEL[101] and manually placed in the electron density. The overall structure was then iteratively rebuilt with COOT[97] and refined in PHENIX[96] to lead to the final structure described in Supplementary Table 1 (PDB:7ly7). There is one molecule in the asymmetric unit and a symmetry mate forms the biological dimer of this complex.

**BmdB–BmdC(Ox)LinkPt and BmdB–BmdC(Ox)LinkBp constructs and assays.** Sequences of linkers used for the BmdB-BmdCLinkPt construct were obtained from an RREL_Ox-embedded NRPS in *Paenibacillus tianmuensis* (WP_090670142.1—https://www.ncbi.nlm.nih.gov/protein/WP_090670142.1?report=genpept; N-terminal linker ISSEEQEIGIPTTQR; C-terminal linker DVKRPEYAAGETVRETAAAVTAASGNR), and those for BmdB-BmdCLinkBp were obtained from an RREL_Ox-embedded NRPS in *Bacillus pseudomycoides* (WP_081621455.1—https://www.ncbi.nlm.nih.gov/protein/WP_081621455.1?report=genpept; N-terminal linker SKNELDLREVSIEKSINKQSKIDK; C-terminal linker KEIESYEEYNQISETENNLKEF). pBacTRev_BmdB was modified by site-directed mutagenesis to insert BamH1 and HindIII sites after the codon for His1535 to make pBacBmdB_BamH1HindIII with primers listed in Supplementary Table 6. Synthetic (Bio Basic Corp) BmdCLinkPt, and BmdCLinkBp sequences in pUC19 were digested by BamH1 and HindIII and ligated into pBacBmdB_-BamH1HindIII. These initial constructs contain the full-length BmdC. The RREL domains of BmdC within these constructs were removed by site-directed mutagenesis primers listed in Supplementary Table 6 to create BmdB–BmdC(Ox)LinkPt and BmdB–BmdC(Ox)LinkBp construct. These mutants were expressed and purified as described for BmdB_S792A.

Reactions performed with these embedded were in a volume of 100 µL of 1 µM BmdB construct, 50 mM HEPES pH 7.5, 100 mM NaCl, 0.5 mM TCEP, 10 mM MgCl$_2$, 1 mM L-alanine, 1 mM L-cysteine, 2 mM ATP, 5 mM tryptamine and 2 mM FMN, incubated for 2 hrs at 37 °C. The reaction was stopped by the addition of 100 µL of 4:1 n-butanol:chloroform. These samples were then lyophilized overnight before resuspension in 50 µL of 10% methanol. Sample (10 µL) was applied to a ZORBAX Extend-C18 (Agilent) column and analyzed with LC-ESI-MS using LC-method 3. Bacillamide production was quantified as described above for assays, including BmdC_D225R.

**IndC purification.** The gene encoding IndC[69,103] from *Streptomyces chromofuscus* (AFV27434.1), which produces indigoidine and consists of the domains A-Ox-PCP-Te, was codon-optimized and synthesized by Bio Basic Corp. and subcloned in the vector pBacT, resulting in the vector pBacT_SC_IndC. IndC was expressed and purified as described for BmdB_S792A.

**EM structural determination and analysis.** BmdB–BmdC complex formation for EM experiments was performed as described above, with pantetheine or Ala-Cys$_{thiazole}$-amino-ppant attached to PCP$_2$. Lacey carbon grids (200 mesh, SPI supplies) were glow discharged for 15 s at 15 mA. Sample (3.5 µl) of were added to the grid surface, blotted for 2 s, and rapidly plunged into liquid ethane, using a Vitrobot (Mark IV, FEI Company). The frozen grids were imaged at 300 kV in a Titan Krios, equipped with a Gatan K3 and Bioquantum (Gatan Inc.) using SerialEM[104]. Each micrograph was acquired in counting mode and consisted of 40 frames to a total fluence of 109 e$^-$/Å$^2$ with a pixel size of 0.855 Å.

All processing steps were carried out in cryoSPARC V2.15 software. All 8140 movies were motion-corrected using patch motion correction. Following patch CTF estimation, blob picking was performed using an elliptical template of size 50 × 300 Å. Only particles with local NCC values between 192 and 521 were kept, resulting in 3,156,357 particles that were extracted using a box size of 512 pixels and binned by 4. Four rounds of 2D classification were performed to remove debris and broken particles, and the remaining 453,440 particles were extracted again using a box size of 640 pixels and downsampled to 480 pixels, resulting in a pixel

size of 1.14 Å. Three ab initio models were generated and used for four rounds of 3D heterogenous refinement as to bad particles. The final particle stack, containing 117,491 particles, was used for nonuniform refinement, resulting in a 4.2 Å reconstruction. The map displayed a full dimer complex, with one BmdB molecule showing much better signal than the other. Next, a mask containing this arm and the entire oxidase dimer was created with Chimera and used for local refinement as nonuniform refinement, resulting in the final 3.8 Å map.

**Reporting summary**. Further information on research design is available in the Nature Research Reporting Summary linked to this article.

## Data availability

The crystallographic data (Supplementary Table 1) used in this study are available in the Protein Data Bank database under accession codes 7ly4, 7ly5, 7ly6, and 7ly7. The cryo-EM models and maps (Supplementary Tables 2 and 3) in this study are available in the Protein Data Bank database under accession codes 7ly4, Electron Microscopy Data Bank under accession codes EMD-23587 and EMD-23588. Source data are provided with this paper.

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

## Acknowledgements

We thank all the members of the Schmeing lab for important advice and ongoing discussions on this project; David Dai for help with cloning; synchrotron staff Shaun Labiuk (Canadian Light Source), and Frank Murphy and Surajit Banerjee (Advanced Photon Source) for facilitating remote collection of diffraction datasets; and staff at the McGill University Facility for EM Research for help with EM data collection. Special thanks go to Dr. Christopher Thibodeaux for many invaluable discussions and his expertise in flavin-dependent enzymes and RiPPs, to Dr. Martin Weigt and Dr. Juan Rodriguez-Rivas at Sorbonne Université for co-evolution discussions and to Nancy Rogerson for proofreading. We are grateful for small molecules from John Colucci and Robert Zamboni (Zamboni Chem Solution) and a gift of small molecules from Jon Patteson and Dr. Bo Li (University of North Carolina Chapel Hill). We thank Christian Chalut (CR-CNRS) for the generous gift of E. coli BL21 (DE3) entD-. This work includes research conducted at the Northeastern Collaborative Access Team beamlines, which are funded by the National Institute of General Medical Sciences from the National Institutes of Health (P30 GM124165). The Eiger 16 M detector on the 24-ID-E beamline is funded by a NIH-ORIP HEI grant (S10OD021527). This research used resources of the Advanced Photon Source, a U.S. Department of Energy (DOE) Office of Science User Facility operated for the DOE Office of Science by Argonne National Laboratory under Contract No. DE-AC02-06CH11357. Data were collected using beamline 08-ID-1 at the Canadian Light Source, a national research facility at the University of Saskatchewan which is supported by the Canada Foundation for Innovation (CFI), the Natural Sciences and Engineering Research Council (NSERC), the National Research Council (NRC), the Canadian Institutes of Health Research (CIHR), the Government of Saskatchewan, and the University of Saskatchewan. This work was funded by CIHR (FDN-148472, 178084) Grants and a Canada Research Chair to TMS, and NSERC MSc and PhD awards to CMF.

## Author contributions

T.M.S. and C.M.F. designed the study and wrote the manuscript. C.M.F. performed biochemical experiments with help from C.H. C.M.F. performed the crystallography. C.M.F., K.B., and T.M.S. determined the structure of BmdC. C.M.F. determined the co-complex crystal structures, with help in refinement from I.S. I.S. and M.S. performed the electron microscopy, data analyses, and map calculations.

## Competing interests

The authors declare no competing interests.
