## [Peer Review File · Nature Communications]

Structures and function of a tailoring oxidase in complex with a nonribosomal peptide synthetase moduleREVIEWER COMMENTS

Reviewer #1 (Remarks to the Author):

Most NRPS biosynthetic pathways include an NRPS and additional proteins that introduce chemical modifications before, during or after assembly-line synthesis. For example, bacillamide biosynthetic pathway is a three-protein BGC that contains genes for BmdA, BmdB, an NRPS and BmdC. BmdC appeared likely to oxidize the thiazoline ring to the thiazole, but there is no indication of the timing of this oxidation. In this study, the bacillamide pathway is reconstituted *in vitro* by the authors. The oxidase is shown to perform dehydrogenation of the thiazoline in the peptide intermediate while it is covalently attached to the NRPS, as the penultimate step in bacillamide D synthesis. Structural analysis of the oxidase using X-ray crystallography and cryo-EM reveals a dimeric, two-lobed architecture with a remnant RiPP recognition element and a dramatic wrapping loop. The oxidase forms a stable complex with the NRPS and dimerizes it. The structures enabled a proof-of-principle bioengineering experiment in which the BmdC oxidase domain is embedded into the NRPS.

The experiments were well designed. It is recommended for publication in NC after minor revision, for example:

1. It should be more specific in the novelty description: this study reveals unexpected biochemical aspects and elegant structural features previously undescribed for any nonribosomal peptide synthesis (lines 77-79)

2. There are many format problems in the references (lines 729-968), which must be adjusted carefully:

(1) some journal names are abbreviated but some are not, e.g. lines 736, 776, 814, 820, 828, 831.....

(2) microbial name must be italic, e.g. *Streptomyces fulvissimus* (line 743), *Pseudomonas aeruginosa* (line 788), *Bacillus* (line 811), *Cyanotheca* (line 842).....

(3) the first letter of the reference title must be low case except the first word, e.g. Identification of Cyclic Depsipeptides and Their Dedicated Synthetase (line 760), Recruitment and Regulation of the Non-ribosomal Peptide Synthetase Modifying Cytochrome P450 Involved in Nikkomycin Biosynthesis (line 781), Nonribosomal Peptide Synthesis-Principles and Prospects (line 792), Switch between One- and Two-electron Chemistry of the Human Flavoprotein Iodotyrosine Deiodinase Is Controlled by Substrate (line 819-820), Synthetase: An Octameric Protein Complex Converting a Ribosomally Synthesized Peptide into a DNA Gyrase Poison (line 824-825), Structural Insights into Thioether Bond Formation in the Biosynthesis of Sactipeptides (line 830-831), and line 846, 864.....

Reviewer #2 (Remarks to the Author):

This manuscript describes the structure and function of an NRPS which synthesizes Bacillamide D. The ternary structure of the complex, the BmdB-BmdC complex, was analyzed by X-ray crystallography and cryo-EM methods. These structures showed that BmdB exists as a BmdC-induced dimer. Most of the experiments are appropriately performed, and the manuscript is clearly written. However, there are several concerns that should be addressed.

Major concerns:

The tertiary structure and biochemical analysis of the BmdB-BmdC complex revealed a part of the functional mechanism of this complex. However, it is still unclear why the oxidation only occurs in

the BmdB-BmdC complex. Although the authors discussed oxidase-NRPS interactions with several examples, there are no discussions to explain the results of Figs 3a, b, and c. It is fantastic if there is experimental evidence to explain these results based on 3D structures. At least, the authors should provide some discussions on this point.

Figure S5a FSC lines are a bit poor. The authors should give some explanations for these FSC lines. Furthermore, they only provided FSC lines, and it is difficult to validate the results of the SPA analysis. In general, SPA analyses have been published with a flow chart of the SPA analysis, angular distribution, FSC curves (corrected maps, masked maps, unmasked maps, phase randomized masked maps), map-to-model FSC curve, and information of local resolutions. Please add these data to the supplementary information.

Minor concerns:

p. 5, line 113 The authors used SEC for analyzing the molecular weight of proteins. However, it is not appropriate to use SEC for the molecular weight analysis. The molecular weight should be analyzed by SEC-MALS. At least, the authors should provide a standard line for SEC with molecular weight markers.

p. 6, line 120 "rmsd values of 2.0-2.9" should be "rmsd values of 2.0-2.9 Å". Furthermore, the number of matched atoms (probably, the number of CA atoms) should be provided.

p. 6, line 141-142 While they described that an FMN cofactor was found in the BmdC, no details are provided about catalytically important atoms in the FMN. They should provide more structural information around the FMN and discuss the catalytic mechanism of oxidation by BmdC.

p.8, line 208 They stated that "with the Ox:A2 interface seen at slightly different angles". It is better to describe this part more quantitatively.

p. 11, line 290 ... is common in oxidases40, 73-75). The last ")" should be removed.

p.11, line 299 ... relative orientations66). The last ")" should be removed.

p. 11, line 302 - p. 12, line 329 This part seems to be a review article. I wonder if these descriptions are required for the manuscript. I think this part should be shortened and clarify the point.

Figure 1b Colors of the EIC and UV lines are very similar. Please use clearer colors. In addition, the font size of the axis title is too small. Particularly, superscripts are too small to see. The font size used in Sup Figure 1 is also tiny. Since molecular weight information is important, the authors should use a larger font size.

Figure 3d There are many lines in one SEC chart, and it is hard to distinguish. Line colors are similar to each other. Please combine different colors and different line styles for clarity (Figure 5d is much better).

Table S1, the column of PDB:7LY7 The statistics said that B-factors of protein atoms and ligand atoms were 178.71 and 99.39 Å², respectively. B-factors for the ligand atoms were substantially smaller than those of the protein atoms. However, it is unlikely. Something wrong seems to be going on in the crystallographic refinement of 7LY7. The authors should check the results of the crystallographic refinement carefully.

Figure S3b Lines in the graphs are difficult to distinguish. Please use clearer colors or use solid and dashed lines to make the differences clear.

Reviewer #3 (Remarks to the Author):

In this manuscript Fortinez et al describe the structural and functional analysis of a tailoring oxidase involved in the synthesis of the nonribosomal peptide, baccillamide D. Tailoring enzymes play an important role in non-ribosomal peptide synthesis but remain poorly characterized. The authors have demonstrated that the oxidase acts on a substrate tethered to the NRPS enzyme and that the oxidase forms a complex in solution with the NRPS enzyme. They have also used X-ray crystallography and Cryo-EM to determine the structure of the oxidase alone and in complex with the 2nd module of the NRPS enzyme. The structural information provides insights into how the oxidation reaction of the tethered peptide by the oxidase is integrated into peptide synthesis by the NRPS. Similar oxidases are found as domains embedded in NRPS enzymes in nature and the authors have used bioengineering to embed the baccillamide oxidase into the NRPS enzyme and shown that the embedded oxidase remains functional. The manuscript is well written and the conclusions are supported by the data. The methodology is sound and while there are clearly some issues remaining with the accuracy of some of the structural models (indicated by relatively high R-factors and number of geometry outliers) they are adequate for the resolution and likely as accurate as possible with the data available. This is a very interesting paper that significantly contributes to the understanding of non-ribosomal peptide synthesis and should be accepted with minor revisions.

A few suggestions and minor criticisms:

Line 176: The authors reference Supplementary Fig. 3c here but I believe they meant to reference 3b iv?

The authors have not described the interaction observed between the A and Cy domains of the BmdB module. I assume it is similar to other NRPS A-C or A-Cy interactions but it would be useful to include a sentence or two to clarify this in the manuscript.

Line 212: At first glance, I found it a bit difficult to understand what the authors meant by "focusing on the BmdC dimer". I suggest rewording and perhaps include a bit more methodological detail here to clarify.

In Fig 4 b The authors have used white to depict the EM structure which is very similar to the colour used for one of the BmdC monomers. I suggest changing the colour of one of these to make the overlay a bit clearer.

Line 260: It appears that BmdC(Ox)LinkPt and BmdB-BmdC(Ox)LinkBp are swapped as it is BmdB-BmdC(Ox)LinkBp that produces total bacillamide at 50% of the wild-type amount.

Response to reviewers

We thank the reviewers for their positive reception to our manuscript and for their insightful comments. We have addressed their points in the revised manuscript, as described in this point-by-point response:

R1.1: It should be more specific in the novelty description: this study reveals unexpected biochemical aspects and elegant structural features previously undescribed for any nonribosomal peptide synthesis (lines 77-79)

A: We have edited this line to be more specific: "... reveals unexpected biochemical aspects, including stable dimerization induced by a tailoring protein, and an elegant NRPS-tailoring protein structure, previously undescribed for any nonribosomal peptide synthesis."

R1.2: There are many format problems in the references (lines 729-968), which must be adjusted carefully: Journal name abbreviations; microbial name italicized; sentence case for titles

A: Apologies, these problems come from downloading references into EndNote from different sources. We have corrected these problems.

R2.1: The tertiary structure and biochemical analysis of the BmdB-BmdC complex revealed a part of the functional mechanism of this complex. However, it is still unclear why the oxidation only occurs in the BmdB-BmdC complex. Although the authors discussed oxidase-NRPS interactions with several examples, there are no discussions to explain the results of Figs 3a, b, and c. It is fantastic if there is experimental evidence to explain these results based on 3D structures. At least, the authors should provide some discussions on this point.

A: All results are consistent with the conclusion that oxidation requires substrate delivery by PCP2, likely because BmdC:PCP interactions are required for productive substrate binding to the Ox active site. The structures show the active site of BmdC to be fairly wide and shallow, hinting that a small molecule would not make extensive interactions, which is also consistent with the need for PCP2:Ox protein-protein interactions for substrate delivery. We now mention this in the discussion.

R2.2: Figure S5a FSC lines are a bit poor. The authors should give some explanations for these FSC lines. Furthermore, they only provided FSC lines, and it is difficult to validate the results of the SPA analysis. In general, SPA analyses have been published with a flow chart of the SPA analysis, angular distribution, FSC curves (corrected maps, masked maps, unmasked maps, phase randomized masked maps), map-to-model FSC curve, and information of local resolutions. Please add these data to the supplementary information.

A: We have added all requested information to Supplemental Fig. 6 and its legend.

R2.3: p. 5, line 113 The authors used SEC for analyzing the molecular weight of proteins. However, it is not appropriate to use SEC for the molecular weight analysis. The molecular

weight should be analyzed by SEC-MALS. At least, the authors should provide a standard line for SEC with molecular weight markers.

A: We agree with R2 that the SEC-MALS is an excellent way of determining molecular weight. However, in this case, we believe that use of SEC is appropriate because it provides the appropriate information on relative molecule weights of species. Furthermore, the interpretation of elution volumes as monomers or dimers of each species is supported by the structural information. We have added our standards to Supplemental Fig. 4 and the indication of elution volume of molecular weight standards to all SEC figures.

R2.3: p. 6, line 120 “rmsd values of 2.0-2.9” should be “rmsd values of 2.0-2.9 Å”. Furthermore, the number of matched atoms (probably, the number of CA atoms) should be provided.

A: Changed to “2.0-2.9 Å over 61-69 alpha carbons”

R2.4: p. 6, line 141-142 While they described that an FMN cofactor was found in the BmdC, no details are provided about catalytically important atoms in the FMN. They should provide more structural information around the FMN and discuss the catalytic mechanism of oxidation by BmdC.

A: We have expanded the passage describing the interactions with FMN and likely mechanism (p6, second paragraph).

R2.5: p.8, line 208 They stated that “with the Ox:A2 interface seen at slightly different angles”. It is better to describe this part more quantitatively.

A: We have specified that the angle varies by around 11 degrees.

R2.6: p. 11, line 290 ... is common in oxidases40, 73-75). The last “)” should be removed.

A: Removed.

R2.7: p.11, line 299 ... relative orientations66). The last “)” should be removed.

A: Removed.

R2.8: p. 11, line 302 - p. 12, line 329 This part seems to be a review article. I wonder if these descriptions are required for the manuscript. I think this part should be shortened and clarify the point.

A: We have shortened this section as requested.

R2.9: Figure 1b Colors of the EIC and UV lines are very similar. Please use clearer colors. In addition, the font size of the axis title is too small. Particularly, superscripts are too small to see. The font size used in Sup Figure 1 is also tiny. Since molecular weight information is important,

the authors should use a larger font size.

A: We have changed the blue color used for IEC lines to better differentiate from the UV lines and increased font size in the axes of Figure 1b and Supplemental Figure 1.

R2.10: Figure 3d There are many lines in one SEC chart, and it is hard to distinguish. Line colors are similar to each other. Please combine different colors and different line styles for clarity (Figure 5d is much better).

A: We have changed line colours and used dashing for some lines.

R2.11: Table S1, the column of PDB:7LY7 The statistics said that B-factors of protein atoms and ligand atoms were 178.71 and 99.39 Å², respectively. B-factors for the ligand atoms were substantially smaller than those of the protein atoms. However, it is unlikely. Something wrong seems to be going on in the crystallographic refinement of 7LY7. The authors should check the results of the crystallographic refinement carefully.

A: Thank you for spotting this. Corrected on re-refinement. The B-factors between ligand and protein are much closer. The ligand B-factors are still somewhat higher because the ligands are physically located closer to the centre of the complex, and the B factors increase at towards the complex periphery.

R2.12: Figure S3b Lines in the graphs are difficult to distinguish. Please use clearer colors or use solid and dashed lines to make the differences clear.

A: We have changed some line colours and used dashing for some lines.

R3.1: Line 176: The authors reference Supplementary Fig. 3c here but I believe they meant to reference 3b iv?

A: Corrected, thank you.

R3.2: The authors have not described the interaction observed between the A and Cy domains of the BmdB module. I assume it is similar to other NRPS A-C or A-Cy interactions but it would be useful to include a sentence or two to clarify this in the manuscript.

A: Added “The Cy-A domain interface observed in BmdB is similar to that seen in FmoA3⁷³ and in other C-A containing structures^{67,74-76}.”

R3.3: Line 212: At first glance, I found it a bit difficult to understand what the authors meant by “focusing on the BmdC dimer”. I suggest rewording and perhaps include a bit more methodological detail here to clarify.

A: Changed to “the highest quality map was obtained by performing refinement with a mask which enveloped the BmdC dimer and a single BmdBM2 protomer”

R3.4: In Fig 4 b The authors have used white to depict the EM structure which is very similar to the colour used for one of the BmdC monomers. I suggest changing the colour of one of these to make the overlay a bit clearer.

A: Changed.

R3.5: Line 260: It appears that BmdC(Ox)LinkPt and BmdB-BmdC(Ox)LinkBp are swapped as it is BmdB-BmdC(Ox)LinkBp that produces total bacillamide at 50% of the wild-type amount.

A: Changed, thank you.

Ed: We strongly encourage you to deposit all new data associated with the paper in a persistent repository where they can be freely and enduringly accessed.... To maximise the reproducibility of research data, we strongly encourage you to provide a file containing the raw data.

A: Structures and maps have been deposited to the PDB and EMDB, and we have now uploaded raw data files containing our underlying data for biochemical experiments.

REVIEWERS' COMMENTS

Reviewer #2 (Remarks to the Author):

The authors have adequately responded to the comments from this reviewer. While I suggested some experiments, such as SEC-MALS, I can accept the authors' response. I think the revised manuscript is improved. Although some original figures were unclear, the revised figures are clearer than before and easy to recognize.

Response to reviewers

We thank the reviewers for the positive reception to the re-submitted manuscript.

R2.1: The authors have adequately responded to the comments from this reviewer. While I suggested some experiments, such as SEC-MALS, I can accept the authors' response. I think the revised manuscript is improved. Although some original figures were unclear, the revised figures are clearer than before and easy to recognize.

A: Thank you.